# Development and Application of Adaptive Evaluation System for TBM Tunneling Based on Case-Based Reasoning

**Jinwu Zhan [1,2], Jiacheng Wang [1], Song Chen [3,4,*], Caisong Luo [1] and Yalai Zhou [1]**

1 School of Civil Engineering, Fujian University of Technology, Fuzhou 350118, China
2 Key Laboratory of Underground Engineering, Fujian Province University, Fuzhou 350118, China
3 College of Urban Geology and Engineering, Hebei Geo University, Shijiazhuang 050031, China
4 Hebei Technology Innovation Center for Intelligent Development and Control of Underground Built Environment, Hebei GEO University, Shijiazhuang 050031, China
* Correspondence: chennsongg@163.com

**Abstract:** The excavation adaptability evaluation decision process for the TBM (tunnel boring machine) in long and large tunnels under complex geological conditions is an uncertain and fuzzy problem affected by multiple factors. Aiming to shift the status quo of TBMs tunneling adaptability evaluation and the ineffective use of related accident cases, the TBM tunneling adaptability evaluation method based on case-based reasoning is proposed and researched. The case-based reasoning method can use existing engineering experience and knowledge to answer newly encountered problems, and can accurately and efficiently evaluate the adaptability of TBM tunneling. Based on the nearest neighbor method, this paper establishes the calculation formula of TBM tunneling adaptability similarity. Based on the statistical analysis method, the evaluation indicators that play a relatively important role in the system are selected, and the TBM tunneling adaptability evaluation index system is constructed. The analytic hierarchy process (AHP) is used to determine the weight of the evaluation indicators at each level. According to the characteristics of TBM tunneling adaptability evaluation, the overall design of the case-based reasoning-based TBM tunneling adaptability evaluation decision system CBR-TBMEAEDS (case-based reasoning-TBM excavation adaptive evaluation decision system) is proposed, and the TBM tunneling adaptability evaluation case is expressed The case-based reasoning method and modification method were designed, and CBR-TBMEAEDS was developed. The system can be used to evaluate the adaptability of TBM to the constructed case library, and the evaluation results are consistent with the actual situation.

**Keywords:** TBM selection; nearest neighbor; case-based reasoning; similarity; decision system

## 1. The Introduction

In recent years, in order to ensure the sustainable and rapid development of China's economy and the implementation of the "the Belt and Road" development strategy, a large number of major tunnel projects, such as water conservancy, railway (highway) and mining, have been built. It is estimated that, in the next 10 years, a number of inter basin water transfer projects, and more than 40 super large hydropower stations, will be built in the western region. The railway (highway) network is being extending to the western region, while projects for extracting mineral resources, such as iron ore and coal mine, are extending to depths of more than 1000 m. The deep-buried long tunnel is the key means of connecting these major infrastructure development projects. Accounting for environmental, topographical and geological conditions, as well as the typical construction period, costs and current technological progress, the TBM (tunnel boring machine) construction method is the primary method of constructing deep and long tunnels [1].

Due to the large depth of the tunnel, the intense tectonic movement means that the surrounding rock presents high stress [2]. At the same time, the intense tectonic movement creates complex and changeable geological conditions in the engineering area; the tunnel

often encounters negative geological bodies, such as fault fracture zone, water-rich fracture zone and mudded interlayer. In addition, adverse geological effects are obvious, including rock burst, water surge, high ground temperature and harmful gases [3–5]. These poor strata present many challenges to TBM tunneling construction. If handled improperly, these challenges can delay the construction period, cause huge economic losses, and even cause casualties in serious cases. For example, the Hongjishi Water Diversion Project was constructed with a double shield (with a diameter of 3.655 m) produced by Robbins Company. The excavation began in November 2008, and experienced 17 instances of blocking both before and after the start of the project. The surrounding rock changes frequently due to high ground stress, large deformation of surrounding rock, serious impact of fault fracture zone, segment damage and shield extrusion damage; the TBM direction is also difficult to control. The main reason for the above construction difficulties was the lack of a systematic and perfect TBM tunneling adaptability evaluation system and associated theory. Therefore, the research on the method of evaluating TBM tunneling adaptability under complex geological conditions has become a major issue; this problem needs to be urgently solved to simplify the construction of long tunnels in the fields of transportation, water conservancy and deep resource development.

To improve the tunneling adaptability of TBM, relevant experts and scholars have also carried out extensive research [6–9]. For example, researchers at the University of Oviedo in Spain [10] have carried out analysis and research on methane emission in single-shield TBM tunneling in Carboniferous strata. Mikaeil et al. [11] used multi-factor fuzzy method to classify the excavatability of TBM under the condition of hard rock. Benardos and Kaliampakos [12] introduced a vulnerability index to describe the uncertainty of formation parameters, and proposed a method to evaluate geological hazards in TBM tunneling. Sapigni et al. [13] obtained the correlation of relevant parameters through a large number of data analyses, and used rock–soil classification standards to evaluate the performance of TBM. Gong et al. [14] discussed the main problems encountered by TBM tunneling under adverse geological conditions, and corresponding treatment measures, in a recent review report. To study the adaptability of EPB shield in expansive clay layer, Song et al. [15] proposed the target design of shield selection, and the selection of excavation parameters, based on the Xuzhou Metro Line 2 phase I project. Lin et al. [16] used analytic hierarchy process (AHP) to construct the TBM adaptability evaluation index system. Based on the membership function, a performance-oriented TBM tunneling adaptability evaluation method is established. Guo et al. [17] proposed a three-stage method to predict the collapse area in real-time by combining the data generated by TBM with the deep learning algorithm. Taking Xinjiang water conservancy project as their background context, Tan et al. [18] determined 11 TBM tunnel adaptability evaluation indexes by considering the influence of tunnel excavation parameters, geological conditions and adverse geological factors on TBM tunnel adaptability.

In addition, due to the complexity of the problem, some mathematical methods that can be used for multi-factor analysis, such as analytic hierarchy process [19], decision analysis [20], expert system [21], have been applied in TBM tunneling adaptability assessment and other aspects. It can be seen from the above literature that, although many studies have been conducted on the tunneling performance and capability of TBM, as factors affecting the tunneling and risks in construction, a set of comprehensive considerations have not yet been formed.

At present, artificial intelligence is being gradually applied to the construction of TBM tunnel [22–24]. Case-based reasoning (CBR) is an intelligent method emerging in the field of artificial intelligence in recent years. It can simulate human thinking to solve problems and reasoning activities, according to People's Daily, and is suitable for solving difficulties related to model building and a lack of domain knowledge [25,26]. In 1982, Professor Schank of Yale University proposed the case-based reasoning cognitive model and its framework for the first time [27]. On the basis of Professor Schank's research, Kolodner and her students developed CYRUS casing-reasoning system [28], marking the moment

at which case-based reasoning officially became a research area in the field of artificial intelligence. At present, this method has been successfully applied in many fields [29–31], but it is rarely applied in TBM adaptability evaluation. It has only been introduced from the principle level, as although the application process is described, the description of key links, such as case representation and retrieval, is too vague. By using CBR thought to establish a TBM-adaptive evaluation and decision system, and rationally using existing TBM tunneling engineering cases, the evaluation and decision results of TBM tunneling can be accurately and efficiently obtained. Therefore, it is a becoming increasing useful to apply AI to TBM tunneling adaptability evaluation. By aiming at "deep", "long", tunneling method with a lack of a reliable evaluation systems, and considering the numerous influencing factors and complex geological conditions of the current TBM project, researchers can use AI theories, methods and technologies to research intelligent decision support systems. These systems can not only greatly promote the construction of TBM tunnel engineering in China, but also avoid the engineering accidents caused by the inadaptability of TBM tunneling; these systems could also make the decision evaluation of TBM tunneling adaptability more efficient, scientific and reasonable.

In this paper, in view of the uncertain and fuzzy problems that are affected by multiple factors in the decision making associated with TBM tunneling adaptability, case-based reasoning-related research on the TBM tunneling adaptability assessment decision making system is carried out. Based on the principle of case-based reasoning, the similarity between the target case and the source case is retrieved according to the attribute and weight value of the case evaluation index; the formula of TBM adaptive similarity calculation is constructed, while the TBM adaptive evaluation index system is determined. The overall design of CBR-TBMEAEDS is presented, including the structure and function design of the system, evaluation and decision process and case-based design. The CBR-TBMEAEDS is developed, and by determining the function of each sub-module of the system, it can achieve the purpose of TBM-adaptive intelligence evaluation and decision; this enables it to have preliminary self-learning ability. Finally, the CBR-TBMEAEDS developed in this paper is verified by a practical TBM engineering case, to evaluate the accuracy and rationality of the system.

## 2. Case-Based Reasoning Method for TBM Tunneling Adaptability

### 2.1. Working Principle of Case-Based Reasoning

CBR is a type of memory derived from cognitive science and reasoning, using actual experience or history to solve existing problems [26,32,33]. It is based on two assumptions: (1) similar or identical problems will have similar or identical solutions, and; (2) similar or identical problems will be encountered repeatedly. Figure 1 shows the flow chart of the CBR working principle. CBR reasoning includes four main steps: (1) case representation; (2) Case retrieval; (3) Case revision, and; (4) Case study.

As a hot research field in artificial intelligence, case-based reasoning (CBR) can use existing engineering experience knowledge to solve new problems. This method is especially useful for TBM tunneling adaptability evaluation, which is affected by multiple, uncertain factors. In addition, there are a large number of past examples of TBM engineering, meaning that case-based reasoning can be used for thorough evaluation. Therefore, this article applies it to the evaluation of TBM tunneling adaptability, which has strong feasibility and value.

### 2.2. Similarity Calculation of TBM Tunneling Adaptability

The main case retrieval methods included the following: nearest neighbor method, inductive reasoning method, knowledge guidance method and template method [34–36]. TBM tunneling adaptability retrieval is based on the principle of retrieving cases with similar characteristics to the source case. In this paper, the nearest neighbor method is used for case retrieval. Its core purpose is to give corresponding weights to the feature attributes of each case. According to the attribute and weight value of case evaluation index, the

similarity between the target case and the source case is retrieved and calculated. Finally, the case with the highest similarity is obtained.

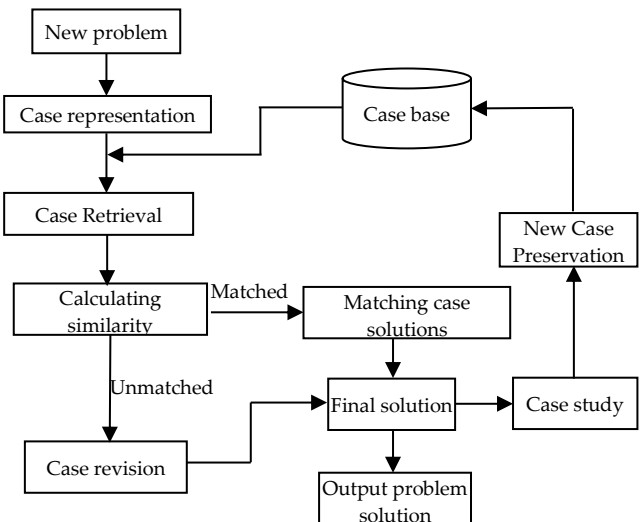

**Figure 1.** Schematic diagram of the CBR working principle.

A data vector **V** usually represents a case. If every case $v_i \in V(i = 1, 2, \cdots, n)$ is represented by a vector $v_i = \{v_{i1}, v_{i2}, \cdots, v_{ik}, \cdots, v_{im}\}$, $n$ is the number of cases in the case space **V**, and $m$ is the dimension of the vector. The foundation of case-based reasoning is a similarity calculation. Successful of case-based reasoning mainly depends on the measurement method and standard of similarity. The dissimilarity between two cases is generally measured by the distance of the characteristic space. Similarity refers to the fact that, when two cases $i$ and $j$ are similar, the value of $sim(i, j)$ is very large; when two cases $i$ and $j$ are not similar, the value of $sim(i, j)$ is very small. The measure standard $sim(i, j)$ of similarity is reflexive.

which is: $sim(i, j) = sim(j, i)$, $\forall i, j \in V$

After standardization, the measure standard of similarity is:

$$0 \leq sim(i, j) \leq 1, \ \forall i, j \in V \tag{1}$$

The dissimilarity is generally called distance, which relates distance to similarity and uses $d(i, j)(\forall i, j \in V)$ to represent this fact. When distance is used to measure the similarity between two cases, if case $i$ and $j$ are similar, the value of distance $d(i, j)$ is very small. If case $i$ and $J$ are not similar, then the distance value $d(i, j)$ is very large.

The metric of distance is symmetric:

$$d(i, j) = d(j, i), \ \forall i, j \in V \tag{2}$$

It also satisfies the triangle inequality:

$$d(i, k) \leq d(i, j) + d(i, k), \ \forall i, j, k \in V \tag{3}$$

In addition, the similarity should meet the following three requirements:

(1) The similarity is non-negative, mean $sim(i, j) \geq 0$.
(2) The similarity of the case itself should be the largest, which is 1.
(3) When the class satisfies compactness, the monotone function of distance between points is similarity.

The distance between the target case feature and the source case feature is calculated by the definition of Equations (2) and (3). The most common measures are the following:

1.  Absolute distance

$$d(i,j) = \sum_{k=1}^{n} \left| v_{ik} - v_{jk} \right| \tag{4}$$

where $v_{ik}$ and $v_{jk}$ are the $k$th attribute value of case $i$ and case $j$, respectively.

2.  Euclidean distance

$$d(i,j) = \sqrt{\sum_{k=1}^{n} \left( v_{ik} - v_{jk} \right)^2} \tag{5}$$

3.  Minkowski distance

$$d(i,j) = \left[ \sum_{k=1}^{n} \left| v_{ik} - v_{jk} \right|^q \right]^{\frac{1}{q}} \tag{6}$$

where $q > 1$. When $q = 1$ and 2, they are absolute distance and Euclidean distance, respectively.

4.  Chebyshev distance

$$d(i,j) = \max_{1 \le k \le q} \left| v_{ik} - v_{jk} \right| \tag{7}$$

When $q \to +\infty$, the limit of Minkowski distance is Chebyshev distance.

The distance obtained above is related to the dimension of each variable index. The influence of dimension can be eliminated through standardized processing of data. The standardized data is:

$$v_{ik}^* = \frac{v_{ik} - \overline{v}_k}{s_k}, i = 1, 2, \cdots, n; \ k = 1, 2, \cdots q \tag{8}$$

among them, $\overline{v}_k = \frac{1}{n} \sum\limits_{i=1}^{n} v_{ik}, s_k^2 = \frac{1}{n-1} \sum\limits_{i=1}^{n} \left( v_{ik} - \overline{v}_k \right)^2$.

With the definition of distance Formulas (4)–(7), we can get the definition of the $k$-th attribute value similarity between two cases:

$$sim_{ij} = 1 - d(i,j), \ \text{when} d(i,j) \in [0, 1] \tag{9}$$

or

$$sim_{ij} = \frac{1}{1 + d(i,j)}, \ \text{when} d(i,j) \in [0, \infty) \tag{10}$$

In fact, each attribute contributes to the overall similarity of a case to different degrees, so the weight value of each attribute should be added. Thus, the formula for calculating the similarity between the two cases of TBM selection can be obtained:

$$sim(i,j) = \frac{\sum\limits_{k=1}^{n} w_k \times sim_{ij}}{\sum\limits_{k=1}^{n} w_k} \tag{11}$$

where $w_k$ is the weight value of the $k$th attribute, $sim_{ij}$ is the similarity value of the $k$-th attribute, and $n$ is the total number of attributes. $sim(i, j)$ is the similarity between the target case and the source case, and $sim(i, j) \in [0, 1]$. When the value is closer to 1, it shows that the two cases are more similar.

## 3. TBM Tunneling Retrieval Characteristic Attributes and Weight Acquisition

### 3.1. Determination of TBM Adaptability Evaluation Indexes

Before using the nearest neighbor method to retrieve cases, it is necessary to determine the feature attributes and their weights. There are many factors that affect the comprehensive evaluation of TBM tunneling adaptation, so various quantitative and uncertain

qualitative factors need to be considered. To evaluate the adaptability of TBM tunneling, the evaluation index, which plays a relatively important role in the system, should be selected; this index reflects the objectives and requirements of TBM tunneling adaptability evaluation, and should be as scientific and comprehensive as possible. Therefore, the evaluation index should be screened.

A large number of engineering practice experiences and research [37–39] show that the factors affecting the adaptability of TBM excavation are mainly divided into five aspects: TBM design, tunnel design, geological conditions, adverse geological problems and construction organization management; these aspects include five criteria layer indexes and 34 index layer indexes. The preliminary TBM driving adaptability evaluation decision-making index system is investigated by experts. Experts are required to divide the importance of each index by 1~10 points, and use statistical software for statistical analysis to further screen the indexes. The mean, standard deviation and coefficient of variation of each index are calculated (Tables 1 and 2). The corresponding mean, screening basis and calculation formulae are shown in Equations (12)–(14):

(1) The mean, which measures the importance of each indicator. The larger the mean value, the higher the relative importance of the evaluation index of the TBM's tunneling adaptability. Only the decision evaluation index whose mean value reaches more than 6 points is retained.

$$M_j = \frac{1}{m}\sum_{i=1}^{n} X_{ij} \tag{12}$$

In the formula, $M_j$ is the $j$-th index mean, $m$ is the total number of experts, $X_{ij}$ is the $i$-th expert and the $j$-th index score.

(2) Standard deviation, reflecting the degree of dispersion of a data set. The greater the standard deviation, the greater the differences between the numerical and mean for the TBM adaptability evaluation index.

$$S_j = \sqrt{\frac{1}{N-1} \times \sum_{i=1}^{N} \left(X_{ij} - M_j\right)^2} \tag{13}$$

In the formula, $S_j$ is the standard deviation of the $j$-th index, and $N$ is the total number of indicators.

(3) Coefficient of variation, reflecting the fluctuation degree of the decision index. The smaller the coefficient of variation, the higher the experts in a certain decision evaluation index of coordination degree. When the coefficient of variation of $V_j < 0.25$, the index's expert coordination degree meets the requirements.

$$V_j = \frac{S_j}{M_j} \tag{14}$$

In the formula, $V_j$ is the variation coefficient of the $j$-th index.

**Table 1.** Statistical analysis results of TBM tunneling adaptive criteria layer indicators.

| Criterion Level Index | Mean | Standard Deviation | Variable Coefficient |
|---|---|---|---|
| TBM design | 7.73 | 0.541 | 0.070 |
| Tunnel design | 6.65 | 0.495 | 0.074 |
| Geologic setting | 8.41 | 0.427 | 0.051 |
| Bad geological problem | 9.17 | 0.516 | 0.056 |
| Organization management in construction | 6.39 | 0.551 | 0.086 |

**Table 2.** Statistical analysis results of TBM excavation adaptive index indicators.

| Criterion Level Index | Index Layer Index | Mean | Standard Deviation | Variable Coefficient |
|---|---|---|---|---|
| TBM design | Type selection design of cutterhead body | 4.24 | 0.514 | 0.121 |
| | Support structure design of cutterhead | 4.36 | 0.714 | 0.164 |
| | Opening rate design | 5.04 | 0.525 | 0.104 |
| | Knife spacing design | 6.71 | 0.693 | 0.103 |
| | Rotating speed of cutter | 6.97 | 0.605 | 0.087 |
| | Rated torque of cutter plate | 8.73 | 0.498 | 0.057 |
| | Rated thrust of cutter plate | 8.71 | 0.733 | 0.084 |
| | Release torque of knife disc | 8.28 | 0.687 | 0.083 |
| | Backup system | 5.51 | 0.602 | 0.109 |
| Tunnel design | Tunnel depth | 8.25 | 0.782 | 0.095 |
| | Tunnel length | 6.33 | 0.679 | 0.107 |
| | Radius of tunnel flat curve | 6.95 | 0.638 | 0.092 |
| | Tunnel gradient | 3.37 | 0.576 | 0.171 |
| | Tunnel section size | 3.61 | 0.563 | 0.156 |
| Geological conditions | Uniaxial compressive strength of rock | 9.32 | 0.281 | 0.030 |
| | Integrity coefficient of rock mass | 9.39 | 0.264 | 0.028 |
| | Volume joint number of rock mass | 4.14 | 0.545 | 0.132 |
| | Rock quality index (RQD) | 4.21 | 0.541 | 0.129 |
| | Quartz content | 7.59 | 0.540 | 0.071 |
| | Abrasion index of rock | 7.62 | 0.498 | 0.065 |
| | Ground stress level | 7.11 | 0.588 | 0.083 |
| | Permeable rate | 6.53 | 0.739 | 0.113 |
| Bad geological problem | Water inrush | 9.47 | 0.424 | 0.045 |
| | Rockburst | 8.65 | 0.540 | 0.062 |
| | Fault fracture zone | 9.04 | 0.289 | 0.032 |
| | Large deformation of surrounding rock under compression | 9.06 | 0.282 | 0.031 |
| | Karst | 7.29 | 0.671 | 0.092 |
| | Compound stratum | 7.74 | 0.636 | 0.082 |
| | High ground temperature | 6.87 | 0.735 | 0.107 |
| | Noxious gas | 6.65 | 0.620 | 0.093 |
| Organization management in construction | Technical level of construction | 7.53 | 0.562 | 0.075 |
| | Construction management level | 6.45 | 0.683 | 0.106 |
| | TBM transportation and assembly | 3.35 | 0.521 | 0.155 |
| | TBM maintenance and disassembly | 3.73 | 0.586 | 0.157 |

It can be seen from Table 1 that the mean values of the five criteria are all greater than six, and the coefficients of variation are all less than 0.25; these findings indicate that the above five criteria are generally recognized by experts, with high concentration and credibility.

It can be seen from Table 2 that 24 of the 34 decision making indexes meet the requirements of mean value greater than six and coefficient of variation less than 0.25; this indicates that the 24 indexes are generally recognized by experts and have high concentration and reliability. The evaluation indexes of TBM tunnelling adaptability selected in this paper are shown in Table 3. Among them, 10 indexes do not meet the screening criteria of the index system and are excluded. The 10 excluded indexes are: the type selection design of cutterhead body, support structure design of cutterhead, opening rate design, backup system, tunnel gradient, tunnel section size, volume joint number of rock mass, rock quality index (RQD), TBM transportation and assembly, TBM maintenance and disassembly.

**Table 3.** Adaptive evaluation indexes of TBM tunnelling.

| Target Layer | Adaptability of TBM Tunneling (D) | | | | |
|---|---|---|---|---|---|
| Rule layer | TBM design $P_1$ | Tunnel design $P_2$ | Geological conditions $P_3$ | Bad geological problem $P_4$ | Organization management in construction $P_5$ |
| Index layer | Rotating speed of cutter $U_1$ Rated torque of cutter plate $U_2$ Rated thrust of cutter plate $U_3$ Release torque of knife disc $U_4$ Knife spacing design $U_5$ | Tunnel depth $U_6$ Tunnel length $U_7$ Radius of tunnel flat curve $U_8$ | Uniaxial compressive strength of rock $U_9$ Integrity coefficient of rock mass $U_{10}$ Quartz content $U_{11}$ Abrasion index of rock $U_{12}$ Ground stress level $U_{13}$ Permeable rate $U_{14}$ | Water inrush $U_{15}$ Rockburst $U_{16}$ Fault fracture zone $U_{17}$ Large deformation of surrounding rock under compression $U_{18}$ Karst $U_{19}$ Compound stratum $U_{20}$ High ground temperature $U_{21}$ Noxious gas $U_{22}$ | Technical level of construction $U_{23}$ Construction management level $U_{24}$ |

### 3.2. Weight Acquisition of TBM Tunneling Adaptability Evaluation Index

As can be seen from Table 3, a large number of evaluation indicators are screened out; some are highly correlated, while others are difficult to quantify. To determine the influence degree of different indicators, combined with TBM tunnel construction experience, existing relevant research results and expert opinions, the judgment matrix is assigned according to the mutual influence of these indicators; the analytic hierarchy process is adopted to determine the weight of the evaluation index. As shown in Table 4, the importance determination scaling method for each factor of the nine scales suggested by Saaty [40] is used. In Table 4, $U_i$ compared to $U_j$ gives $a_{ij}$, and $U_j$ compared to $U_i$ gives $a_{ji} = 1/a_{ij}$. The judgment matrix of each level is constructed by Table 4, and the eigenvector and eigenroot of the matrix are obtained by the method of sum. After completing the consistency test, each component of the eigenvector corresponding to the consistency requirements is the weight of each evaluation index (Tables 5–10).

**Table 4.** Nine scales for the importance scale method of each factor.

| $a_{ij}$ | Basis |
|---|---|
| 1 | $U_i$ is just as important as $U_j$ |
| 3 | $U_i$ is slightly more important than $U_j$ |
| 5 | $U_i$ is significantly more important than $U_j$ |
| 7 | $U_i$ is more important than $U_j$ |
| 9 | $U_i$ is more important than $U_j$ |
| 2, 4, 6, 8 | The median values of the above two judgments |

**Table 5.** Influence weight analysis of criterion layer evaluation index.

| D | $P_1$ | $P_2$ | $P_3$ | $P_4$ | $P_5$ |
|---|---|---|---|---|---|
| $P_1$ | 1 | 5 | 1/2 | 1/3 | 7 |
| $P_2$ | 1/5 | 1 | 1/6 | 1/7 | 3 |
| $P_3$ | 2 | 6 | 1 | 1/2 | 8 |
| $P_4$ | 3 | 7 | 2 | 1 | 9 |
| $P_5$ | 1/7 | 1/3 | 1/8 | 1/9 | 1 |
| Single-layer weight | 0.1890 | 0.0568 | 0.2881 | 0.4353 | 0.0307 |
| $\lambda_{\max} = 5.1837$, $CI = 0.0459$, $RI = 1.1185$, $CR = 0.0411 < 0.1$ | | | | | |

**Table 6.** Influence weight analysis of P1 index layer evaluation indexes.

| $P_1$ | $U_1$ | $U_2$ | $U_3$ | $U_4$ | $U_5$ |
|---|---|---|---|---|---|
| $U_1$ | 1 | 1/4 | 1/4 | 1/3 | 2 |
| $U_2$ | 4 | 1 | 1 | 2 | 5 |
| $U_3$ | 4 | 1 | 1 | 2 | 5 |
| $U_4$ | 3 | 1/2 | 1/2 | 1 | 4 |
| $U_5$ | 1/2 | 1/5 | 1/5 | 1/4 | 1 |
| Single-layer weight | 0.0840 | 0.3317 | 0.3317 | 0.1976 | 0.0550 |
| $\lambda_{\max} = 5.0553, CI = 0.0138, RI = 1.1185, CR = 0.0124 < 0.1$ | | | | | |

**Table 7.** Influence weight analysis of P2 index layer evaluation index.

| $P_2$ | $U_6$ | $U_7$ | $U_8$ |
|---|---|---|---|
| $U_6$ | 1 | 7 | 3 |
| $U_7$ | 1/7 | 1 | 1/5 |
| $U_8$ | 1/3 | 5 | 1 |
| Single-layer weight | 0.6491 | 0.0719 | 0.2790 |
| $\lambda_{\max} = 3.0649, CI = 0.0324, RI = 0.5149, CR = 0.0630 < 0.1$ | | | |

**Table 8.** Influence weight analysis of P3 index layer evaluation indexes.

| $P_3$ | $U_9$ | $U_{10}$ | $U_{11}$ | $U_{12}$ | $U_{13}$ | $U_{14}$ |
|---|---|---|---|---|---|---|
| $U_9$ | 1 | 1 | 5 | 5 | 6 | 8 |
| $U_{10}$ | 1 | 1 | 5 | 5 | 6 | 8 |
| $U_{11}$ | 1/5 | 1/5 | 1 | 1 | 2 | 4 |
| $U_{12}$ | 1/5 | 1/5 | 1 | 1 | 2 | 4 |
| $U_{13}$ | 1/6 | 1/6 | 1/2 | 1/2 | 1 | 3 |
| $U_{14}$ | 1/8 | 1/8 | 1/4 | 1/4 | 1/3 | 1 |
| Single-layer weight | 0.3638 | 0.3638 | 0.0923 | 0.0923 | 0.0585 | 0.0293 |
| $\lambda_{\max} = 6.1697, CI = 0.0339, RI = 1.2494, CR = 0.0272 < 0.1$ | | | | | | |

**Table 9.** Influence weight analysis of P4 index layer evaluation indexes.

| $P_4$ | $U_{15}$ | $U_{16}$ | $U_{17}$ | $U_{18}$ | $U_{19}$ | $U_{20}$ | $U_{21}$ | $U_{22}$ |
|---|---|---|---|---|---|---|---|---|
| $U_{15}$ | 1 | 3 | 2 | 2 | 6 | 5 | 7 | 8 |
| $U_{16}$ | 1/3 | 1 | 1/2 | 1/2 | 4 | 3 | 5 | 6 |
| $U_{17}$ | 1/2 | 2 | 1 | 1 | 5 | 4 | 6 | 7 |
| $U_{18}$ | 1/2 | 2 | 1 | 1 | 5 | 4 | 6 | 7 |
| $U_{19}$ | 1/6 | 1/4 | 1/5 | 1/5 | 1 | 1/2 | 2 | 3 |
| $U_{20}$ | 1/5 | 1/3 | 1/4 | 1/4 | 2 | 1 | 3 | 4 |
| $U_{21}$ | 1/7 | 1/5 | 1/6 | 1/6 | 1/2 | 1/3 | 1 | 2 |
| $U_{22}$ | 1/8 | 1/6 | 1/7 | 1/7 | 1/3 | 1/4 | 1/2 | 1 |
| Single-layer weight | 0.3002 | 0.1330 | 0.2018 | 0.2018 | 0.0448 | 0.0652 | 0.0311 | 0.0221 |
| $\lambda_{\max} = 8.2636, CI = 0.0377, RI = 1.4200, CR = 0.0265 < 0.1$ | | | | | | | | |

**Table 10.** Influence weight analysis of P5 index layer evaluation indexes.

| $P_5$ | $U_{23}$ | $U_{24}$ |
|---|---|---|
| $U_{23}$ | 1 | 4 |
| $U_{24}$ | 1/4 | 1 |
| Single-layer weight | 0.8 | 0.2 |
| $\lambda_{max} = 2.0$, $CI = 0$, $RI = 0$, $CR = 0 < 0.1$ | | |

According to the weight values of the above evaluation indexes, the overall evaluation index weights of TBM tunneling are calculated (Table 11).

**Table 11.** Weight of TBM total level evaluation index.

| The Criterion Level (Local Weights) | The Index Level (Local Weights) | Global Weights |
|---|---|---|
| $P_1$ (0.1890) | $U_1$ (0.0840) | 0.0159 |
| | $U_2$ (0.3317) | 0.0627 |
| | $U_3$ (0.3317) | 0.0627 |
| | $U_4$ (0.1976) | 0.0373 |
| | $U_5$ (0.0550) | 0.0104 |
| $P_2$ (0.0568) | $U_6$ (0.6491) | 0.037 |
| | $U_7$ (0.0719) | 0.0041 |
| | $U_8$ (0.2790) | 0.0158 |
| $P_3$ (0.2881) | $U_9$ (0.3638) | 0.1048 |
| | $U_{10}$ (0.3638) | 0.1048 |
| | $U_{11}$ (0.0923) | 0.0266 |
| | $U_{12}$ (0.0923) | 0.0266 |
| | $U_{13}$ (0.0585) | 0.0169 |
| | $U_{14}$ (0.0293) | 0.0084 |
| $P_4$ (0.4353) | $U_{15}$ (0.3002) | 0.1307 |
| | $U_{16}$ (0.1330) | 0.058 |
| | $U_{17}$ (0.2018) | 0.0878 |
| | $U_{18}$ (0.2018) | 0.0878 |
| | $U_{19}$ (0.0448) | 0.0195 |
| | $U_{20}$ (0.0652) | 0.0284 |
| | $U_{21}$ (0.0311) | 0.0135 |
| | $U_{22}$ (0.0221) | 0.0096 |
| $P_5$ (0.0307) | $U_{23}$ (0.8) | 0.0246 |
| | $U_{24}$ (0.2) | 0.0061 |

## 4. Overall Design of CBR-TBMEAEDS

### 4.1. The Structure and Function Design of the System

According to the characteristics of TBM tunneling adaptability evaluation, the structural design of CBR-TBMEAEDS is proposed (Figure 2). Menu functions mainly include case library management, update case tree, case retrieval, exit system menu functions, etc. Through the case base management menu function, the case base and case table can be created, added and deleted; the case can be updated by updating the menu function of case tree. The case retrieval menu function can also be used to complete case retrieval, sort similar cases, and select the best case; CBR can be terminated by the exit system menu function.

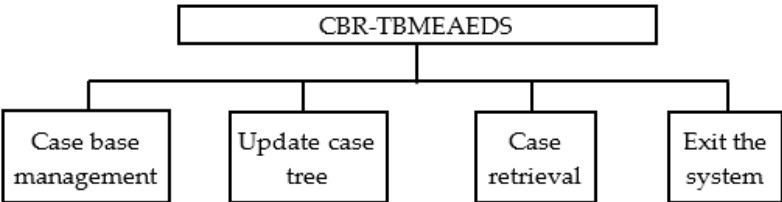

**Figure 2.** Design of CBR-TBMEAEDS's structure and function.

### 4.2. The Evaluation and Decision Process of the System

Figure 3 shows the decision making process of TBM adaptability evaluation, based on case-based reasoning as used in this paper. When evaluating a new TBM case, a large number of existing TBM tunneling adaptability cases in the case base are used to search for the same or similar source cases through case-based reasoning system CBR retrieval; this ensures that decisions made on the TBM tunneling adaptability can be evaluated, and the evaluation results of TBM adaptability can be obtained quickly and accurately. The core of case-based reasoning is to build a case base, which is composed of many cases; each case is composed of corresponding various attribute indicators. New solutions can be obtained by modifying and supplementing past cases, which can adapt to the purpose of new similar cases.

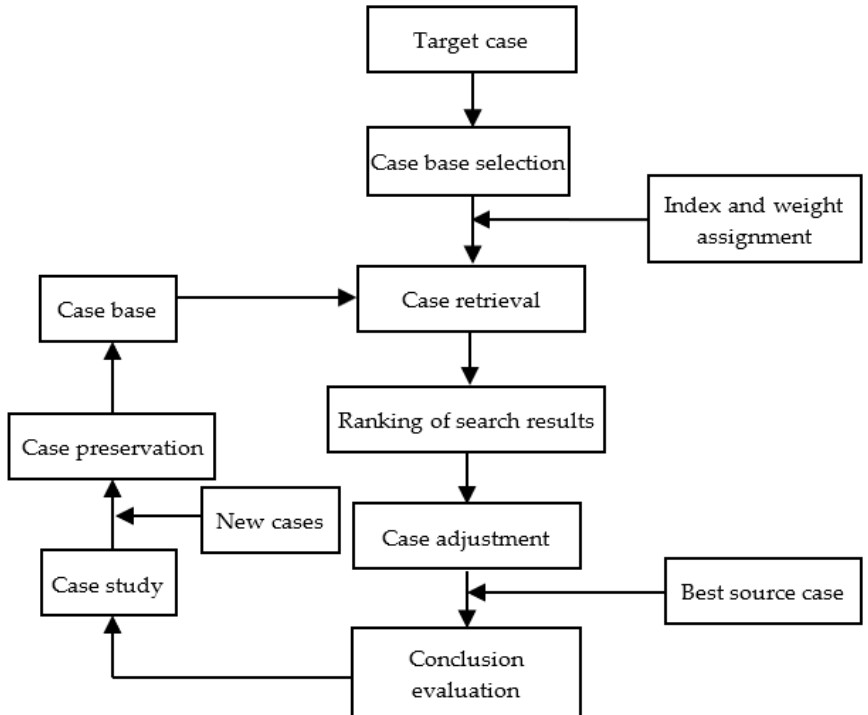

**Figure 3.** Decision making process of CBR-TBMEAEDS.

The specific evaluation and decision making process is as follows: (1) through the interface of the system, input the target case of the problem to be solved, and select the corresponding case base to assign the index and weight; (2) facing the user's input request, the case retrieval module calculates the similarity between the target case and the source case in the case knowledge base, before filtering the source cases, sorting them according to the similarity, and outputting the retrieval results to find the similar cases; (3) the similar cases obtained by retrieval are adjusted and reused, the target cases are compared with the similar cases obtained by reasoning, and the solutions are adjusted according to the current environment of specific evaluation problems, and; (4) after case revision, the new solution

is returned to the user as the result. After case learning, the new case is added to the case base to continuously improve the case base.

The CBR-TBMEAEDS case library is used to store the evaluated TBM cases. In addition, the corresponding indicators of TBM were compiled through Equations (1)–(11); after completing the writing of the system's reasoning rules, the obtained cases were written into the case base, with the the reasoning of the cases completed through the case-based reasoning based TBM tunneling adaptability evaluation and decision system.

The reasoning machine is the control center of the CBR-TBMEAEDS, which resembles the human brain. It includes the reasoning method, strategy and solving process. The inference machine matches the rule knowledge in the knowledge base according to the user input data. The reasoning process is implemented according to the control strategy of the reasoning tree, and the adaptability evaluation of TBM is completed. The model of the evaluation decision making system is shown in Figure 4.

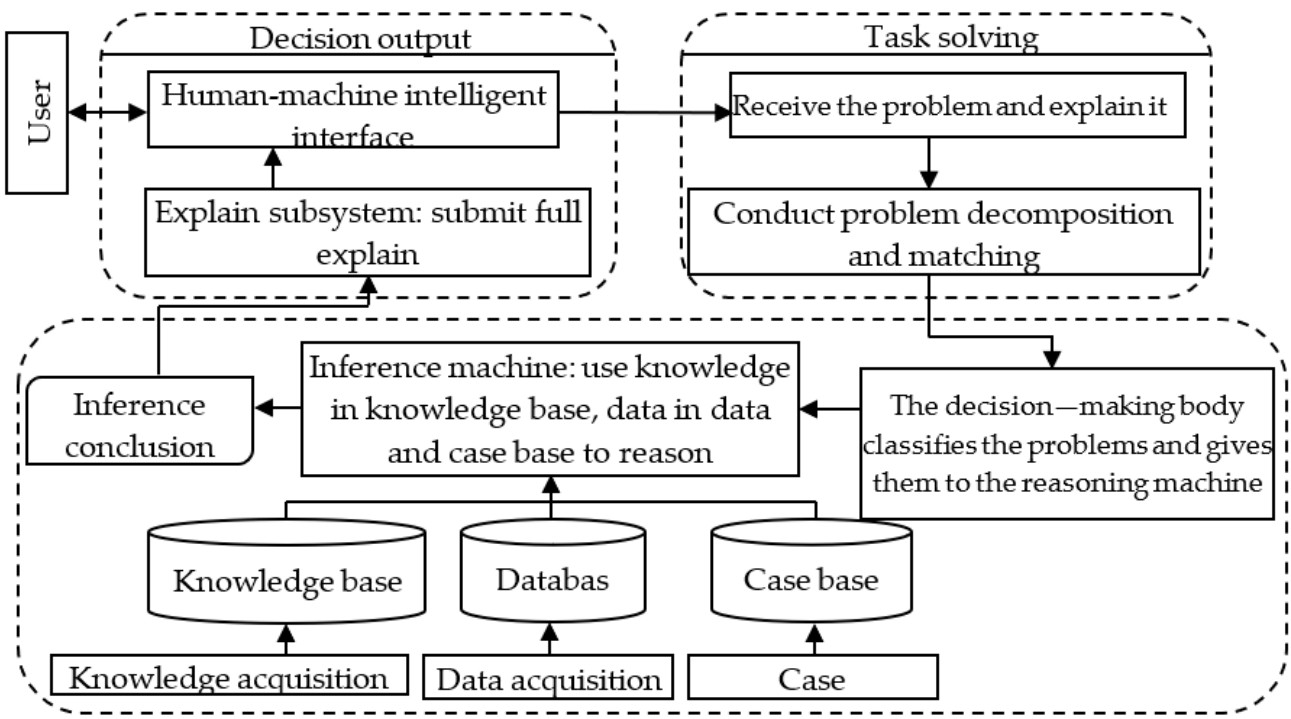

**Figure 4.** Model of the integrated evaluation decision support system.

*4.3. TBM Tunneling Adaptability Case Library Design*

The data of case base is stored in the case table, which mainly provides the source data of case-based reasoning for the system. The system completes the similarity calculation by sourcing the retrieval information in the case table, and obtaining the reasoning result. Table 12 shows an example of the organization of the TBM tunneling adaptability evaluation case library, including the case number, name, evaluation index, adaptability, evaluation level and weight, and other fields. The evaluation index and corresponding weight are the main retrievals of case reasoning information.

**Table 12.** Example of TBM tunneling adaptability evaluation case base organization.

| Case Number | Case Name | Evaluation Index | | Adaptability | Evaluation Level | | Weights |
|---|---|---|---|---|---|---|---|
| | | TBM tunneling evaluation index | | | TBM tunneling evaluation index weight | | |
| | | $U_1$ | Rotating speed of cutter | | $W_1$ | Rotating speed of cutter | |
| | | $U_2$ | Rated torque of cutter plate | | $W_2$ | Rated torque of cutter plate | |
| | | ...... | ...... | | ...... | ...... | |
| | | $U_{23}$ | Technical level of construction | | $W_{23}$ | Technical level of construction | |
| | | $U_{24}$ | Construction management level | | $W_{24}$ | Construction management level | |

## 5. Programming Implementation of CBR-TBMEAEDS

### 5.1. Programmatic Realization of the System

Based on the overall design of the system, the Java language is used for development, and the programming of CBR-TBMEAEDS is realized (Figure 5). As shown in Figure 5, the main interface displays two menu items that can be operated, namely "case base management" and "about the system"; four tool menus are also displayed, namely "new case table", "update case tree", "case retrieval" and "exit the system". The case tree is located on the left-hand side of the interface, the case base and case table in the case tree are displayed according to the tree structure, and the case data in the case table is displayed on the right-hand side.

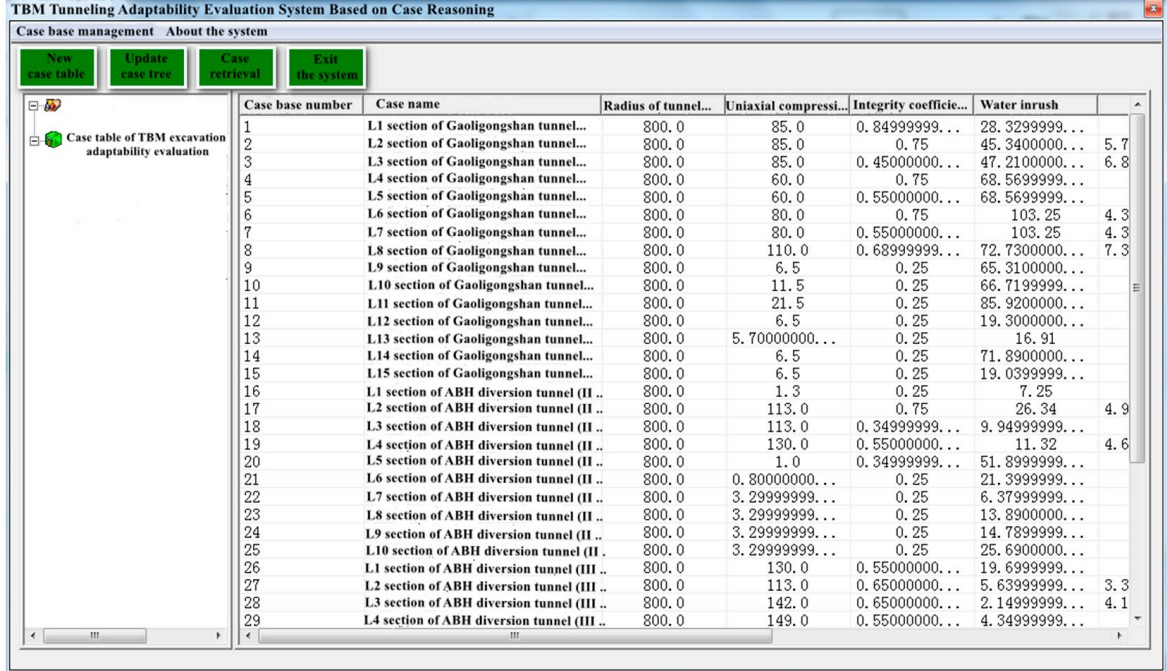

**Figure 5.** Main interface of CBR-TBMEAEDS.

### 5.2. Functions of the System

In this paper, 41 TBM tunneling adaptability evaluation cases are collected, and the TBM driving adaptability evaluation case table is constructed. Figure 6 shows the established TBM tunneling adaptability evaluation case table. A case is made up of a record,

and a relational database is used to build the database. The knowledge representation of each TBM driving adaptability evaluation case should have the following contents: (1) Case number; (2) Case name; (3) Case evaluation index; (4) Adaptability; (5) Adaptability evaluation grade.

| Case base number | Case name | Radius of tunnel... | Uniaxial compressi... | Integrity coefficie... | Water inrush | |
|---|---|---|---|---|---|---|
| 1 | L1 section of Gaoligongshan tunnel... | 800.0 | 85.0 | 0.84999999... | 28.3299999... | |
| 2 | L2 section of Gaoligongshan tunnel... | 800.0 | 85.0 | 0.75 | 45.3400000... | 5.7 |
| 3 | L3 section of Gaoligongshan tunnel... | 800.0 | 85.0 | 0.45000000... | 47.2100000... | 6.8 |
| 4 | L4 section of Gaoligongshan tunnel... | 800.0 | 60.0 | 0.75 | 68.5699999... | |
| 5 | L5 section of Gaoligongshan tunnel... | 800.0 | 60.0 | 0.55000000... | 68.5699999... | |
| 6 | L6 section of Gaoligongshan tunnel... | 800.0 | 80.0 | 0.75 | 103.25 | 4.3 |
| 7 | L7 section of Gaoligongshan tunnel... | 800.0 | 80.0 | 0.55000000... | 103.25 | 4.3 |
| 8 | L8 section of Gaoligongshan tunnel... | 800.0 | 110.0 | 0.68999999... | 72.7300000... | 7.3 |
| 9 | L9 section of Gaoligongshan tunnel... | 800.0 | 6.5 | 0.25 | 65.3100000... | |
| 10 | L10 section of Gaoligongshan tunnel... | 800.0 | 11.5 | 0.25 | 66.7199999... | |
| 11 | L11 section of Gaoligongshan tunnel... | 800.0 | 21.5 | 0.25 | 85.9200000... | |
| 12 | L12 section of Gaoligongshan tunnel... | 800.0 | 6.5 | 0.25 | 19.3000000... | |
| 13 | L13 section of Gaoligongshan tunnel... | 800.0 | 5.70000000... | 0.25 | 16.91 | |
| 14 | L14 section of Gaoligongshan tunnel... | 800.0 | 6.5 | 0.25 | 71.8900000... | |
| 15 | L15 section of Gaoligongshan tunnel... | 800.0 | 6.5 | 0.25 | 19.0399999... | |
| 16 | L1 section of ABH diversion tunnel (II .. | 800.0 | 1.3 | 0.25 | 7.25 | |
| 17 | L2 section of ABH diversion tunnel (II .. | 800.0 | 113.0 | 0.75 | 26.34 | 4.9 |
| 18 | L3 section of ABH diversion tunnel (II .. | 800.0 | 113.0 | 0.34999999... | 9.94999999... | |
| 19 | L4 section of ABH diversion tunnel (II .. | 800.0 | 130.0 | 0.55000000... | 11.32 | 4.6 |
| 20 | L5 section of ABH diversion tunnel (II .. | 800.0 | 1.0 | 0.34999999... | 51.8999999... | |
| 21 | L6 section of ABH diversion tunnel (II .. | 800.0 | 0.80000000... | 0.25 | 21.3999999... | |
| 22 | L7 section of ABH diversion tunnel (II .. | 800.0 | 3.29999999... | 0.25 | 6.37999999... | |
| 23 | L8 section of ABH diversion tunnel (II .. | 800.0 | 3.29999999... | 0.25 | 13.8900000... | |
| 24 | L9 section of ABH diversion tunnel (II .. | 800.0 | 3.29999999... | 0.25 | 14.7899999... | |
| 25 | L10 section of ABH diversion tunnel (II . | 800.0 | 3.29999999... | 0.25 | 25.6900000... | |
| 26 | L1 section of ABH diversion tunnel (III .. | 800.0 | 130.0 | 0.55000000... | 19.6999999... | |
| 27 | L2 section of ABH diversion tunnel (III .. | 800.0 | 113.0 | 0.65000000... | 5.63999999... | 3.3 |
| 28 | L3 section of ABH diversion tunnel (III .. | 800.0 | 142.0 | 0.65000000... | 2.14999999... | 4.1 |
| 29 | L4 section of ABH diversion tunnel (III .. | 800.0 | 149.0 | 0.55000000... | 4.34999999... | |

**Figure 6.** Case table of TBM excavation adaptability evaluation.

In the case retrieval interface, users enter the retrieval value of TBM project demonstration case evaluation index and its corresponding feature weight, and click the [search] button to conduct case retrieval (Figure 7). Figure 8 shows the top 10 case records with the highest similarity.

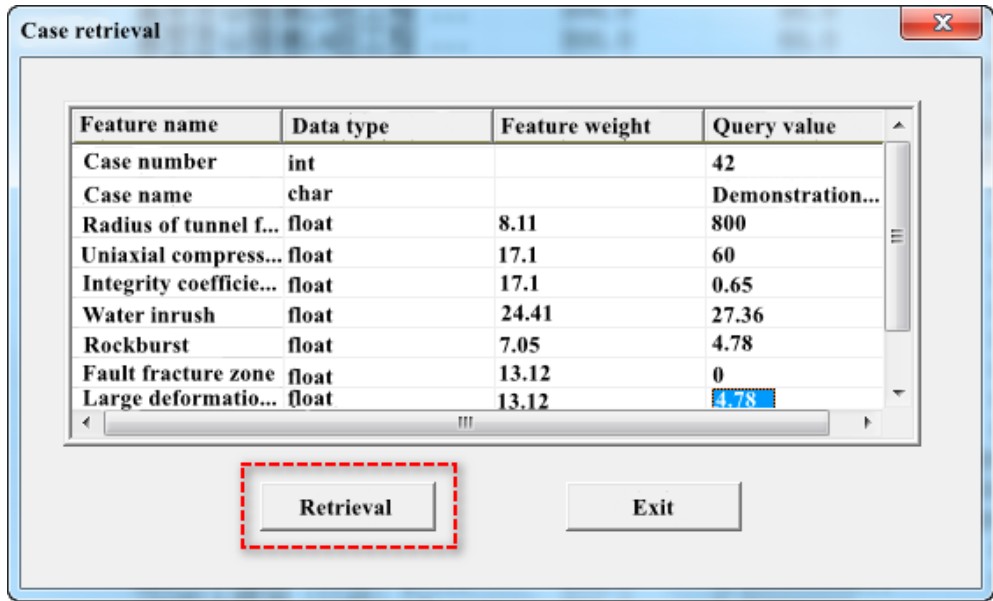

**Figure 7.** Demonstration case retrieval.

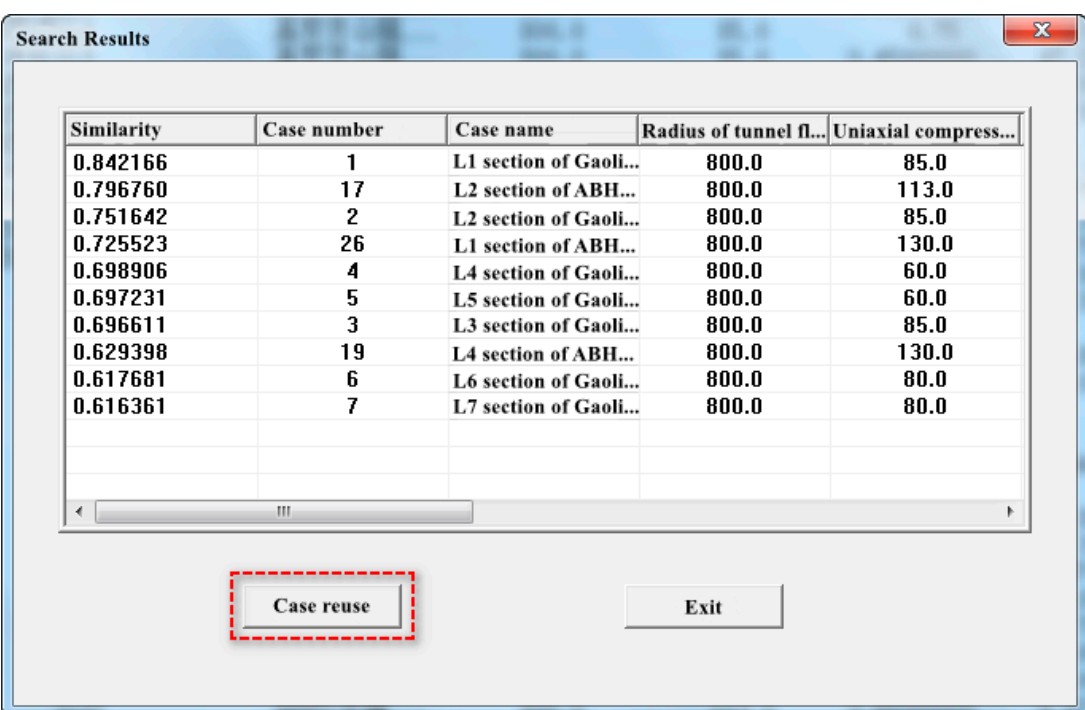

**Figure 8.** Case retrieval results.

Users can either choose a case with the highest similarity for reuse, or choose a more appropriate case for reuse. The user selects an appropriate case, and clicks the "case reuse" button to reuse the case.

When reusing TBM cases, the characteristics of current cases and reuse cases are not necessarily the same. At this time, the corresponding countermeasures are adjusted according to the actual needs of the case. Through the learning function of cases, new cases are added continuously to improve the case base,; new cases help it to accumulate more experience, and make the system have preliminary self-learning ability.

## 6. Application of Engineering Cases

In order to test the applicability of the CBR-TBMEAEDS constructed in this paper, four cases are selected from the case table as the target cases of engineering application, and the tunneling adaptability of TBM is evaluated.

### 6.1. Case Based Reasoning of TBM Tunneling Adaptability

As shown in Table 13, there are four target cases extracted from the TBM excavation adaptability evaluation case base. Users can select the TBM tunneling evaluation case table on the system interface and click case retrieval (Figure 7). Users can then input the target case of the Gaoligongshan tunnel L5 section project into the case retrieval interface (Figure 9) for case reasoning.

**Table 13.** Target cases of TBM tunneling adaptability evaluation.

| Case Name<br><br>Evaluation Index | L5 Section of the Gaoligongshan Tunnel (Case Number 5) | L12 Section of the Gaoligongshan Tunnel (Case Number 12) | L5 Section of the ABH Diversion Tunnel (III Standard) (Case Number 30) | L14 Section of the ABH Diversion Tunnel (III Standard) (Case Number 39) |
|---|---|---|---|---|
| $U_1$ | 6.5 | 6.5 | 9.8 | 9.8 |
| $U_2$ | 11,797 | 11,797 | 4510 | 4510 |
| $U_3$ | 25,133 | 25,133 | 20,040 | 20,040 |

**Table 13.** *Cont.*

| Evaluation Index | L5 Section of the Gaoligongshan Tunnel (Case Number 5) | L12 Section of the Gaoligongshan Tunnel (Case Number 12) | L5 Section of the ABH Diversion Tunnel (III Standard) (Case Number 30) | L14 Section of the ABH Diversion Tunnel (III Standard) (Case Number 39) |
|---|---|---|---|---|
| $U_4$ | 17,695 | 17,695 | 10,045 | 10,045 |
| $U_5$ | 80 | 80 | 83 | 83 |
| $U_6$ | 595 | 687 | 1191 | 1312 |
| $U_7$ | 13.26 | 13.26 | 14 | 14 |
| $U_8$ | 800 | 800 | 800 | 800 |
| $U_9$ | 60 | 6.5 | 149 | 3.4 |
| $U_{10}$ | 0.55 | 0.25 | 0.55 | 0.35 |
| $U_{11}$ | 40.25 | 40.25 | 17.9 | 17.9 |
| $U_{12}$ | 3.34 | 3.34 | 2.2 | 2.2 |
| $U_{13}$ | 0.212 | 3846 | 0.273 | 11.765 |
| $U_{14}$ | 7.37 | 18.59 | 0.05 | 6.06 |
| $U_{15}$ | 68.57 | 19.3 | 3.01 | 10.62 |
| $U_{16}$ | 4.71 | 7 | 3.663 | 7 |
| $U_{17}$ | 0 | 50 | 0 | 7 |
| $U_{18}$ | 4.71 | 0.26 | 3.663 | 0.085 |
| $U_{19}$ | 0.67 | 0.67 | 0.92 | 0.92 |
| $U_{20}$ | 0 | 100 | 0 | 100 |
| $U_{21}$ | 28 | 28 | 32 | 33 |
| $U_{22}$ | 0 | 0 | 0 | 0 |
| $U_{23}$ | 8.5 | 8.5 | 8 | 8 |
| $U_{24}$ | 8.5 | 8.5 | 8 | 8 |

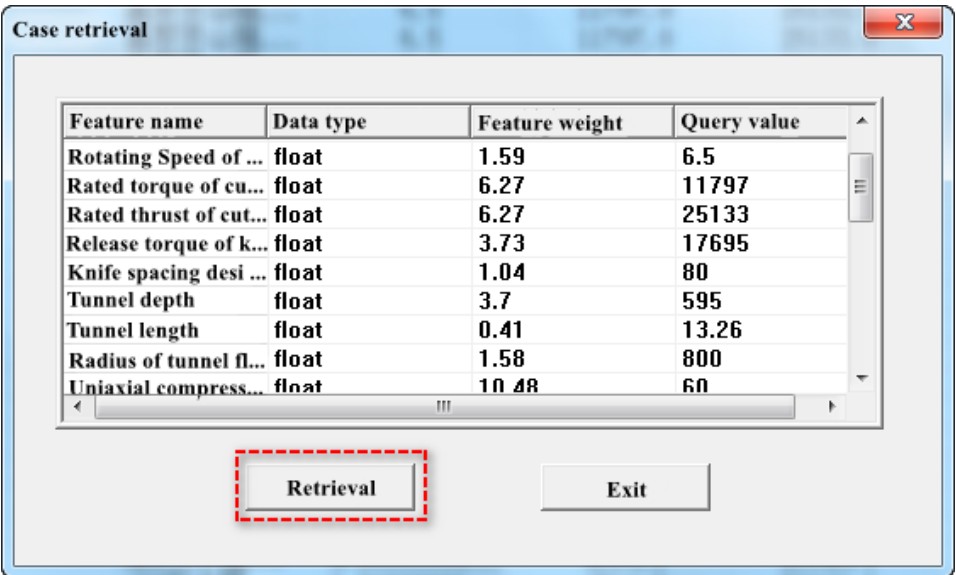

**Figure 9.** Input engineering retrieval information of excavation for Section L5 of the Gaoligongshan tunnel.

The excavation search results for the L5 Section of Gaoligongshan tunnel are obtained (Figure 10), and the top 10 records with the highest similarity to the target case are shown in Figure 10. The first record is the retrieval result of the target case itself, so the similarity is 1; the highest similarity with the target case is the L4 section of Gaoligongshan tunnel with No. 4, and the similarity is 0.904167.

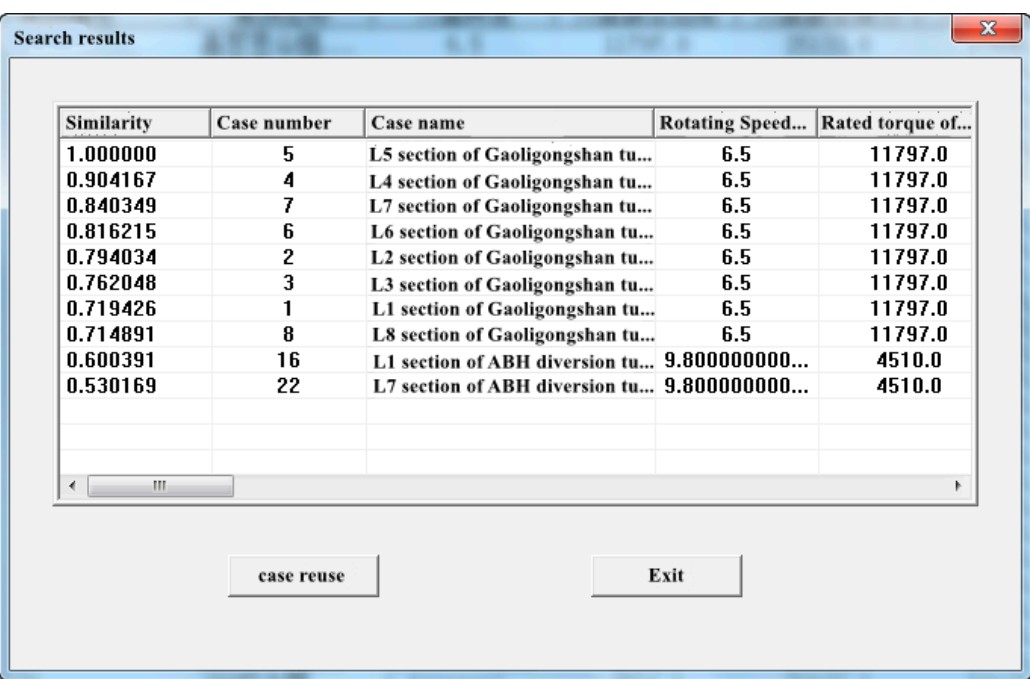

**Figure 10.** Retrieval results of selection for Section L12 of the Gaoligongshan tunnel.

*6.2. Reasoning Result Analysis*

In the same way, the reasoning results of the other target cases can be obtained. Table 14 shows the reasoning results of the target case of the TBM tunneling adaptability evaluation.

From the results of the above-listed TBM tunneling adaptive case reasoning, it can be seen that the TBM tunneling evaluation results based on case-based reasoning basically conform to the following rules: (1) the higher the similarity, the closer the adaptability and adaptability level of both the target case and the source case; (2) the lower the similarity, the greater the difference between the adaptability and adaptability level of the target case and the source case.

The results of case-based reasoning show that: the evaluation results of the target case and the source case are positively correlated with the similarity; that is, the higher the similarity, the closer the evaluation results are.; conversely, a greater te deviation indicates that CBR-TBMEAEDS can carry out effective case-based reasoning. In the follow-up cases, the case base can be continuously improved through the case adjustment and case learning functions of the system, making the evaluation results increasingly accurate and instilling preliminary self-learning ability.

**Table 14.** Reasoning results of TBM sexcavation evaluation target cases.

| Case Number | Target Case | Source Case | Similarity | Target Case Adaptability | | Source Case Adaptability | | Deviation |
|---|---|---|---|---|---|---|---|---|
| | | | | Adaptability | Adaptability Level | Adaptability | Adaptability Level | |
| 5 | L5 section of the Gaoligongshan tunnel | L4 section of the Gaoligongshan tunnel | 0.9042 | 0.8237 | Adaptable (height) | 0.8235 | Adaptable (height) | 0.02% |
| 12 | L12 section of the Gaoligongshan tunnel | L15 section of the Gaoligongshan tunnel | 0.9835 | 0.5624 | Slightly adaptable | 0.5559 | Slightly adaptable | 1.16% |
| 30 | L5 section of the ABH diversion tunnel (III standard) | L4 section of the ABH diversion tunnel (III standard) | 0.6633 | 0.7007 | Adaptable (moderate) | 0.9087 | Completely adaptable | 29.68% |
| 39 | L14 section of the ABH diversion tunnel (III standard) | L11 section of the ABH diversion tunnel (III standard) | 0.8363 | 0.5818 | Slightly adaptable | 0.5836 | Slightly adaptable | 0.31% |

## 7. Conclusions

Using relevant existing research as a basis for study, this paper comprehensively analyzes the nature and characteristics of adaptability factors, such as geological conditions, adverse geological processes and tunnel design; the paper also carries out research on TBM tunneling adaptability evaluation methods based on case-based reasoning. The main work and research results are as follows:

(1) According to the engineering characteristics of TBM tunneling adaptability evaluation, the nearest neighbor method is used to retrieve the cases, and the calculation formula of TBM tunneling adaptability similarity is obtained. Through the analysis of the main influencing factors of TBM excavation adaptability, the characteristic attributes of retrieval and their weights were determined, and the TBM excavation adaptability evaluation index system was constructed.

(2) Based on the CBR method of case-based reasoning, the structure, function design and evaluation decision making process of CBR-TBMEAEDS are proposed, the TBM tunneling adaptability case base is designed, CBR-TBMEAEDS is developed and the function of the system is determined.

(3) From the application results of adaptive case reasoning in TBM tunneling, it can be seen that the evaluation results of the target case and the source case are positively correlated with the similarity. When the similarity reaches 0.8 or more, the fitness is relatively close, and the deviation is 0.02~1.16%. The adaptability levels are the same; when the similarity is less than 0.7, the deviation value is higher, the fitness difference is large, and the adaptability levels are basically inconsistent. That is to say, the higher the similarity between the two, the smaller the deviation value and the closer the evaluation result. In contrast, a greater the deviation value shows the effectiveness of CBR-TBMEAEDS case reasoning.

(4) The application of case-based reasoning in TBM tunneling adaptability evaluation is still in the exploratory stage, and there are still many problems to be solved. This concept has not been verified by a large number of cases, so a more perfect case base should be built to make the evaluation results increasingly accurate. At the same time, case adjustment and learning methods need to be further explored to improve the model's initial self-learning ability.

**Author Contributions:** Conceptualization: J.Z.; methodology: J.Z. and S.C.; software: J.Z. and J.W.; validation: J.Z. and S.C.; Data curation: J.W. and C.L.; writing—original draft preparation, J.W. and C.L.; writing—review and editing: S.C. and Y.Z. All authors have read and agreed to the published version of the manuscript.

**Funding:** The research described in this paper was financially supported by the National Natural Science Foundation of China (No. 52008111), and Natural Science Foundation of Fujian province (No. 2020J01885, No. 2022J05185). This research was funded by Science and Technology Project of Hebei Education Department (QN2023060), and National Pre-research Funds of Hebei GEO University in 2023 (KY202305).

**Institutional Review Board Statement:** Not applicable.

**Informed Consent Statement:** Not applicable.

**Data Availability Statement:** The data that support the findings of this study are available from the corresponding authors, upon reasonable request.

**Conflicts of Interest:** The authors declare no conflict of interest.

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
