# Peer review of "Development and Application of Adaptive Evaluation System for TBM Tunneling Based on Case-Based Reasoning"

_sustainability, doi:10.3390/su15075768_

Round 1
Reviewer 1 Report
1. TBM tunneling adaptability evaluation of long tunnel under complex geological conditions is studied using the relevant accident cases. TBM tunneling adaptability evaluation method based on case-based reasoning is presented. The design of CBR-TBMEAEDS is presented. Some useful conclusions are reached.
2. In literature review part, more detailed information corresponding to the study and background should be presented.
3. Some of the words are covered by the frame line in Fig. 1. It should be revised.
4. In the three main retrieval methods for cases, why the nearest neighbor method is selected in the study.
5. The formula (1)-(11) are suggested to be cited in references.
6. Please present the resources of the results in Tables 1-2. Do the results belong to the authors? How many samples are used in the results? "The preliminary TBM driving adaptability evaluation decision-making index system is investigated by experts." What is the relation between the authors and the experts?
7. In Tables 4-9, the determination of the value of judgment matrix is suggested to be presented in more details.
8. The flow chart and algorithm in the programming of CBR-TBMEA are suggested to be added.
9. I wonder if the Tables 12-14 can be merged in one table.
10. What it the TBM excavation adaptability evaluation case base? Please explain it.
11. The language should be improved under the help of an English native speaker.
Author Response
upload it as a pdf file.

Reviewer 2 Report
The paper proposed a TBM tunneling adaptability evaluation method based on case-based reasoning, which has certain engineering significance and practical value. However, there are points where revisions are required, some minor and others of greater importance.
1. Some figures are not clear, especially, the text is incomplete, like Fig. 1, Fig. 3
2. What’s the difference between I, J and i, j in line 143 ?
3. How are the mean values in Table 1 and Table 2 determined? The method should be explained.
4. The format of references should be rechecked.
Author Response
upload it as a pdf file.

Reviewer 3 Report
Many thanks for submitting the paper entitled "Development and Application of Adaptive Evaluation system for TBM tunnelling based on Case-Based reasoning".
In this paper, the research on the TBM tunnelling adaptability assessment decision-making system based on case-based reasoning is carried out and a new platform was developed to achieve this scope.
Generally, the English language results are deficient in places. Some expressions adopted in the current work by the authors are not suitable for a scientific paper. In other cases, the repetition of entire words in the text or the table is recognized.
Specifically, the following aspects are relieved by the reviewer:
- Text is full of typos. Pay attention to the correct placement of spaces, especially around some punctuation marks;
- Add references as requested in the note added in the revised manuscript;
- improve the quality of figures;
Other specific suggestions are reported in the revised manuscript by the reviewer.
Generally, I think that the paper develops a novel operative platform with an interesting scope. However, as reported in the manuscript by the authors, the proposed method "has not been verified by a large number of cases, so a more perfect case base should be built to make the evaluation results more and more accurate".
For this reasons, I suggest to improve the accuracy of the evaluation method or try to point out a reasonable analysis of the problems which occurs in other cases.
In this form, the submitted paper could require extensive work in order to improve the quality of the written text and the overall merit of the research.
For this reason, I think that the paper cannot be eligible for publication.
Author Response
upload it as a pdf file.

Round 2
Reviewer 1 Report
I think the authors have responded my comments. I suggest that it can be accepted.
Author Response
Thank you again for your valuable comments.
Reviewer 3 Report
Though the authors replied by adding in the cover letter some considerations with specific regards to the overall merit of the paper, several corrections or suggestions reported in the original pdf manuscript were ignored.
Please, revise the document by considering the corrections added directly on the old version of the manuscript during the first review of the original manuscript that you can find in the attachment

Author Response
The authors have made modifications in Word and attached modification instructions. Thank you again for your valuable comments.

Round 3
Reviewer 3 Report
Dear authors,
the revised manuscript can be accepted in the current form